# Genomic alterations involved in fluoroquinolone resistance development in *Staphylococcus aureus*

Thuc Quyen Huynh[1,2,3☯], Van Nhi Tran[1,3☯], Van Chi Thai[1,3], Hoang An Nguyen[1,3], Ngoc Thuy Giang Nguyen[1,3], Minh Khang Tran[1,3], Thi Phuong Truc Nguyen[1,3], Cat Anh Le[1,3], Le Thanh Ngan Ho[1,3], Navenaah Udaya Surian[4], Swaine Chen[4], Thi Thu Hoai Nguyen[1,2,3]*

1 School of Biotechnology, International University, Ho Chi Minh City, Vietnam, 2 Research Center for Infectious Diseases, International University, Ho Chi Minh City, Vietnam, 3 Vietnam National University, Ho Chi Minh City, Vietnam, 4 Genome Institute of Singapore, Singapore, Singapore

☯ These authors contributed equally to this work.
* ntthoai@hcmiu.edu.vn

**Data Availability Statement:** All relevant data are within the paper and its Supporting information files.

## Abstract

### Aim

Fluoroquinolone (FQ) is a potent antibiotic class. However, resistance to this class emerges quickly which hinders its application. In this study, mechanisms leading to the emergence of multidrug-resistant (MDR) *Staphylococcus aureus* (*S. aureus*) strains under FQ exposure were investigated.

### Methodology

*S. aureus* ATCC 29213 was serially exposed to ciprofloxacin (CIP), ofloxacin (OFL), or levofloxacin (LEV) at sub-minimum inhibitory concentrations (sub-MICs) for 12 days to obtain *S. aureus*-1 strains and antibiotic-free cultured for another 10 days to obtain *S. aureus*-2 strains. The whole genome (WGS) and target sequencing were applied to analyze genomic alterations; and RT-qPCR was used to access the expressions of efflux-related genes, alternative sigma factors, and genes involved in FQ resistance.

### Results

A strong and irreversible increase of MICs was observed in all applied FQs (32 to 128 times) in all *S. aureus*-1 and remained 16 to 32 times in all *S. aureus*-2. WGS indicated 10 noticeable mutations occurring in all FQ-exposed *S. aureus* including 2 insdel mutations in SACOL0573 and *rimI*; a synonymous mutation in *hslO*; and 7 missense mutations located in an untranslated region. GrlA, was found mutated (R570H) in all *S. aureus*-1 and -2. Genes encoding for efflux pumps and their regulator (*norA*, *norB*, *norC*, and *mgrA*); alternative sigma factors (*sigB* and *sigS*); acetyltransferase (*rimI*); methicillin resistance (*fmtB*); and hypothetical protein BJI72_0645 were overexpressed in FQ-exposed strains.

**Funding:** This study was financially supported by The Youth Incubator for Science and Technology Programme, managed by Youth Promotion Science and Technology Center - Ho Chi Minh Communist Youth Union and Department of Science and Technology of Ho Chi Minh City, under the contract number 34/2022/ H-KHCNT-VU. The funders had no role in study design, data collection and analysis, decision to publish, or preparation of the manuscript.

**Competing interests:** The authors have declared that no competing interests exist.

## Conclusion

The emergence of MDR *S. aureus* was associated with the mutations in the FQ-target sequences and the overexpression of efflux pump systems and their regulators.

## Introduction

*Staphylococcus aureus (S. aureus)* is a common extracellular pathogen with the ability to cause a wide range of infections ranging from mild (skin infections, and soft tissues) to serious (life-threatening infections, such as pneumonia, osteomyelitis, endocarditis, toxic shock syndromes. . .) [1]. The rates of multidrug resistance in *S. aureus* have been increasing considerably in recent years, leading to serious consequences, including treatment failure, treatment cost consumption, and protracted therapy [2, 3].

Fluoroquinolone (FQ) is a unique synthetic broad-spectrum antibiotics class that has been applied in the treatment of a wide variety of community and nosocomial infections [4]. Unfortunately, the broad use of these drugs has led to a steadily increasing number of FQ-resistant bacterial strains, with rates of resistance varying with both organisms and geographic regions [5]. The proportion of Gram-positive cocci resistance to these drugs increased worldwide, especially for *S. aureus* [6]. Clinical isolates were found to resist to FQs via the modification of the FQ targets, overexpression of efflux pumps, plasmid-mediated resistance, and decrease in permeability due to alterations in the outer membrane [7–11]. Furthermore, other phenotypic characteristics such as the biofilm-forming ability or the presence of small colony variants can also markedly increase the concentrations of FQs required to inhibit and eradicate biofilm compared to planktonic cells [12, 13]. However, the mechanisms leading to the emergence of FQ resistance have not been well understood. It can emerge from gene acquisition, gene mutation, or just simply gene regulation.

It has been speculated that in the presence of antibiotics, bacteria undergo microevolution via the change in their genetic information to adapt to the environment and ensure their survival [14, 15]. This process happens in several ways. Bacteria may acquire genes including drug-resistant genes, through gene transfer processes [16, 17]. Besides, they can become resistant through stress-induced mutagenesis [18–20], in which the bacteria can increase the mutation rate to enhance the probability of mutations that permit adaptations to the stressor. Antibiotics can impose strong selection pressure on microbial populations that keep resistant-gene-carrying individuals surviving, replicating, and quickly becoming the dominant type throughout the microbial population [21–23].

The resistance of *S. aureus* to FQs results from mutations in quinolone resistance-determining regions (QRDRs) of topoisomerase IV and/or DNA gyrase [9, 24]. These enzymes are both tetrameric with pairs of two different subunits: GyrA and GyrB for gyrase and ParC and ParE for topoisomerase IV which are chromosomally encoded by *gyrA*, *gyrB*, *parC*, and *parE*, respectively. The most common mutations of the QRDR include S84 and D87 for GyrA, or S80 and E84 for ParC [25, 26]. In *S. aureus*, genes encoding for topoisomerase IV are named *grlA* (approximately 1992 bp) and *grlB* (approximately 2493 bp) which are analogous to *parC* and *parE* in other species, respectively [25, 27–29]. It is assumed that some of the specific mutations in the topoisomerase IV and DNA gyrase gene identified from clinical strains were involved in FQ resistance [26, 30–32]. Although these alterations appear to decrease their affinity to FQs [33], it is difficult to analyze the contribution of each mutation to the resistance phenotype.

In addition, FQ resistance is also mediated by the overexpression of endogenous efflux systems that pump drugs out of the cell could significantly reduce the accumulation of drugs inside the bacterial cells [34, 35]. Among chromosomal efflux proteins in *S. aureus*, NorA, NorB, and NorC which are encoded by *norA*, *norB*, and *norC* respectively, played an important role in decreased susceptibility to antibiotics especially FQs [36–38]. These proteins are belonged to the major facilitator superfamily (MFS) and negatively regulated by MgrA [37].

In this study, the *in vitro*-induced FQ resistance in the fully susceptible *S. aureus* ATCC 29213 was generated for investigating the mechanism associated with intrinsic FQ resistance development. Whole genome sequencing of *in vitro*-induced FQ-resistant *S. aureus* strains as well as target sequencing of important genes were analyzed to associate the mutations in the target enzymes with the obtained resistant phenotypes. Additionally, the expression of efflux-related genes, alternative sigma factors, and genes involved in FQ resistance were evaluated using RT-qPCR.

## Methods

### Selection of FQ-resistant strains

*S. aureus* ATCC 29213 (initial *S. aureus*), a fully susceptible strain, was cultured in Mueller-Hinton Broth (MHB) that contained sub-minimum inhibitory concentrations (sub-MICs) values of FQ which is either ciprofloxacin (CIP), ofloxacin (OFL) or levofloxacin (LEV) following a previously published procedure [39]. The experiment was repeated until no increase in MIC to the FQ used in exposure was observed. At the endpoint, the obtained CIP-, LEV-, and OFL-exposed *S. aureus* (*S. aureus*-1) strains were sent to NK Biotechnology Company (HCM, Vietnam) for *16S rRNA* sequencing. These selected resistant strains were sub-cultured for another 10 days in an antibiotic-free medium with the daily examination of MIC in order to obtain CIP-, LEV-, and OFL-revertant *S. aureus* (*S. aureus*-2) strains. Repetitive sequence-based PCR (Rep-PCR) amplification was established to confirm the genetic relatedness of initial *S. aureus* ATCC 29213 and those of selected strains. The test was adapted from Vito *et al.* [40]. The strains at day 12th of FQ exposure (*S. aureus*-1 including CIP-, OFL- and LEV-1) and the strains at day 10th of antibiotic-free culture (*S. aureus*-2 including CIP-, OFL- and LEV-2) were then used for other experiments.

### Antimicrobial susceptibility testing

The micro-dilution method on 96-well plates, instructed by EUCAST guidelines (Eucast.org, version 11.0), was applied on seven bacterial strains (initial *S. aureus*, *S. aureus*-1 and -2) to determine the MIC value of the strains to CIP, LEV, OFL, moxifloxacin, nalidixic acid, ampicillin, amoxicillin, chloramphenicol, cefalexin, doxycycline, erythromycin, lincomycin, oxacillin, and tetracycline. At the same time, MICs of the strains were also measured under the presence of an efflux pump inhibitor, reserpine (Sigma-Aldrich, USA), at a concentration of 20 mg/L.

### Whole genome sequencing

DNA extraction was carried out using the GeneJET Genomic DNA Purification Kit (Thermo Scientific, USA) according to the manufacturer's instructions. Sequencing libraries were prepared using the TruSeq Nano DNA LT Library Prep Kit (Illumina, Singapore). Pooled libraries were then sequenced using an Illumina NextSeq 500 sequencer with 2 x 151 bp reads. The resulting FASTQ files were mapped to the ATCC 29213 genome (Genbank GCF_001879295.1) using bwa (version 0.7.10) (PMID 19451168); indel realignment and SNP

(single nucleotide polymorphism) calling were performed using Lofreq* (version 2.1.2) with default parameters (PMID 23066108).

## Quinolone-resistance determining regions (QRDRs) target sequencing

The genomic DNA of *S. aureus* strains (Initial *S. aureus* ATCC 29213, *S. aureus* strains at day 4th, 6th, 8th, 10th of CIP-, LEV- and OFL-exposure, CIP-, LEV-, OFL-resistant strain-1 and -2) were extracted and then, utilized for QRDRs (DNA gyrase—*gyrA* and *gyrB*, and topoisomerase IV—*grlA* and *grlB*) amplification. Each 50 µl PCR reaction was prepared with instructions from the *TopTaq* Master Mix Kit (Qiagen, Germany). The amplification profiles and primers were described previously [41]. The PCR products were finally electrophoresed and subjected to sequencing (Macrogen, Korea).

## Expression evaluation of efflux-related genes, alternative sigma factors, and genes involved in FQ resistance

Total RNA extraction and cDNA Synthesis of seven *S. aureus* strains were performed using Monarch® Total RNA Miniprep Kit (NEB, UK) and SensiFAST cDNA Synthesis Kit (Bioline, UK), according to the manufacturer's instructions. Each real-time qRT-PCR reaction consisted of 20 µL of reagents, including Luna® Universal qPCR Master Mix (NEB, UK), one primer pair (Table 1; Reference gene: *16S rRNA*), cDNA template, and nuclease-free water. The thermal cycle of the RT-qPCR machine was set up based on NEB #M3003 instruction manual (NEB, UK). Transcription values (Ct) are analyzed as described in [42].

## Statistical analysis

The RT-qPCR experiments were performed in triplicates. After obtaining the transcriptional values (Ct) from amplicon-based fluorescence thresholds, the Ct values of the target genes were normalized to that of the *16S rRNA* transcripts to obtain a ΔCt. The $2^{-\Delta\Delta Ct}$ method was then used to compare the relative expression patterns between FQ-exposed strains and the initial *S. aureus* [42]. Fold change was calculated against initial *S. aureus* and visualized on a log scale, with gene expression of initial *S. aureus* as 1. Values greater than one were considered up-regulated, while values smaller than 1 were down-regulated. Fold change and confidence level 95% CI (error bar) were calculated in MS Excel according to the standard practice [43]. Expression data of three biological replicates were analyzed by one-tail Student's t-test to identify the statistical significance of differential expression between FQ-exposed *S. aureus* strains

**Table 1. Primers for qRT-PCR analysis.**

| Genes | Forward primer (5'→3') | Reverse primer (5'→3') | Amplicon Size (bp) | References |
|---|---|---|---|---|
| *mgrA* | GGGATGAATCTCCTGTAAACG | TTGATCGACTTCGGAACG | 131 | [44] |
| *norA* | AATGCCTGGTGTGACAGGTT | TCCACCAATCCCTGGTCCTA | 246 | [45] |
| *norB* | AGCGCGTTGTCTATCTTTCC | GCAGGTGGTCTTGCTGATAA | 213 | [45] |
| *norC* | AATGGGTTCTAAGCGACCAA | ATACCTGAAGCAACGCCAAC | 216 | [45] |
| *rimI* | ATTGCGTCCTCACCTTCACC | CTGAGGCGGAACGAAATTGG | 453 | This study |
| *fmtB* | ACTGCTGTTGCTAATTGTTGA | GCACAAGTTGATGAAGCGAA | 191 | This study |
| Gene encoding hypothetical *protein* BJI72_0645 | GCGAGATGTCCGCTAAAAGT | TGGTGCATGTGATGACGTTG | 191 | This study |
| *sigB* | ATGTACGTTTATTGAAGGATTG | TAATTTCTTAATTGCCGTTCTC | 103 | [46] |
| *sigS* | ACCTTGAAGGATACAAGCAA | GGCATTTACGCTTAACGGAC | 96 | [47] |
| *16S rRNA* | AGAGTTTGATCMTGGCTCAG | GWATTACCGCGGCKGCTG | 492 | [45] |

and the initial *S. aureus* ATCC 29213. A p-value of $\leq 0.05$ was considered statistically significant.

## Results

### Serial exposure to FQs leading to FQ-resistant phenotype

After being serially exposed to FQs, *S. aureus* ATCC 29213 turned from FQ-sensitive to FQ-resistant phenotype. The MIC values of the FQ-exposed strains reached peaks at day 8th-10th of exposure (Fig 1), and no further increase in MIC values was observed after the peaks even if these strains were continuously exposed to the antibiotics. After 12-day serial exposure to FQs, the MIC values of obtained *S. aureus*-1 were 128 times higher than that of the initial strain for CIP, and 32 times higher for OFL and LEV. Impressively, *S. aureus*-2 strains did not revert to antibiotic-sensitive phenotype after being cultured in an antibiotic-free medium for 10 days (remained 8–32 times higher than the initial strain). Gram staining, *16S rRNA sequencing*, and Rep-PCR fingerprint results proved no contamination during the antibiotic exposure process (S1 Fig). In addition, molecular genotyping results showed that *S. aureus*-1 and -2 strains were identical to the initial *S. aureus* ATCC 29213.

### Susceptibility of *S. aureus* strains to different antibiotics

The FQ-exposed *S. aureus* ATCC 29213 turned into a multidrug-resistant phenotype, which was not only resistant to other FQs but also to other antibiotics of unrelated groups (Table 2). The increases in MICs of *S. aureus*-1 and -2 strains were as followings: ampicillin (8–16, 4–16 times), amoxicillin (64–128 times), cefalexin (4–128 times), doxycycline (16–32 times), erythromycin (64–128 times), lincomycin (64–256, 64–128 times) and oxacillin (16–64, 16–32 times). Interestingly, the exposed strains remained sensitive to moxifloxacin (0.25–0.5 mg/L) and unchanged to chloramphenicol and tetracycline. Subsequent cultures of *S. aureus*-1 strains in an antibiotic-free medium only resulted in minor effects on their susceptibility. Most MIC values of *S. aureus*-2 strains decreased only 2 to 4 times compared to *S. aureus*-1 strains. Under the presence of reserpine, *S. aureus* ATCC 29213 was not affected while susceptibility of *S. aureus*-1 strains was reduced by 2–8 folds for some antibiotics such as ampicillin, amoxicillin, doxycycline, erythromycin, and lincomycin (S1 Table).

### Whole genome sequencing of *S. aureus* ATCC 29213, *S. aureus*-1 and *S. aureus*-2 strains

Whole genome sequencing was used to identify mutations associated with the changes in resistance (S2 Table). Overall, a total of 42 positions where at least one strain carried a variant relative to the initial strain were discovered. Of these 42 variant positions, 10 were common in all strains sequenced, indicating differences in our parental strain from the publicly available genome sequence. Among 32 remaining variant positions, 24 were SNPs, 6 were deletions, and 2 were insertions. A total of 11 variants were seen in all 6 of the *S. aureus*-1 and *S. aureus*-2 strains relative to our re-sequenced ATCC 29213 (Table 3). Of these 11 SNPs, 4 were found in predicted protein-coding genes: an amino acid deletion in SACOL0573 (encoding a PIN domain-containing protein); an amino acid insertion in *rimI* (an alanine acetyltransferase); a missense mutation in *grlA*, encoding DNA topoisomerase IV subunit A, and a synonymous mutation in *hslO*, which encodes heat shock protein Hsp33. The remaining 7 variants located in the upstream region of *norA* and gene coding hypothetical protein BJI72_0645 were not found in annotated protein-coding sequences.

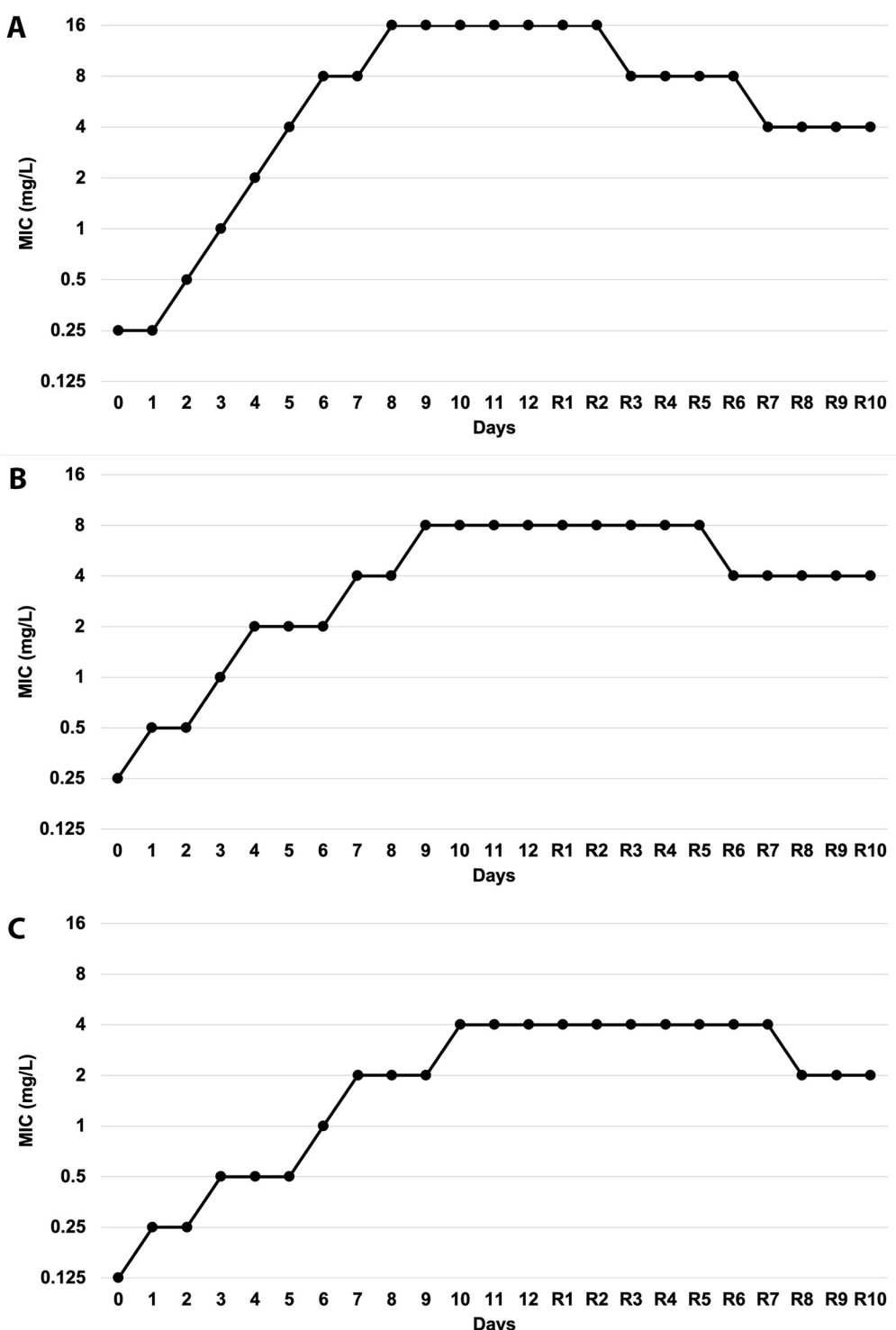

**Fig 1. MIC values of *S. aureus* during sub-MIC exposure to FQs including ciprofloxacin (CIP) (A), ofloxacin (OFL) (B), or levofloxacin (LEV) (C).** The initial *S. aureus* was exposed to CIP, OFL, or LEV for 12 days to obtain FQ-resistant *S. aureus* strains (CIP-1, OFL-1, and LEV-1). These exposed *S. aureus* strains were then continuously sub-cultured for another 10 days in an antibiotic-free medium to obtain CIP-2, OFL-2, and LEV-2 with the daily examination of MICs values evaluated. Day 0: MIC; day 1–12: 12 FQ exposed days; R1-R10: 10 days in antibiotic-free medium.

**Table 2. Antibiotic susceptibility profile of *S. aureus* ATCC 29213 (initial strain), *S. aureus*-1 (CIP-1, OFL-1, and LEV-1) and *S. aureus*-2 (CIP-2, OFL-2, and LEV-2).**

| Antibiotics | *S. aureus* strains | | | | | | |
|---|---|---|---|---|---|---|---|
| | ATCC 29213 | CIP-1 | CIP-2 | LEV-1 | LEV-2 | OFL-1 | OFL-2 |
| *Fluoroquinolone* | MIC (mg/L) | | | | | | |
| Ciprofloxacin | 0.125 | 16 | 4 | 8 | 2 | 4 | 2 |
| Levofloxacin | 0.125 | 4 | 1 | 4 | 2 | 8 | 4 |
| Ofloxacin | 0.25 | 4 | 2 | 16 | 4 | 8 | 4 |
| Moxifloxacin | 0.0625 | 0.25 | 0.25 | 0.5 | 0.5 | 0.5 | 0.25 |
| Nalidixic acid | 16 | 128 | 64 | 128 | 64 | 128 | 64 |
| *Other antibiotics* | | | | | | | |
| Ampicillin | 16 | 256 | 256 | 128 | 64 | 128 | 64 |
| Amoxicillin | 0.5 | 32 | 32 | 64 | 32 | 64 | 64 |
| Cefalexin | 0.5 | 4 | 4 | 64 | 64 | 8 | 8 |
| Chloramphenicol | 16 | 32 | 32 | 16 | 16 | 32 | 16 |
| Doxycycline | 0.125 | 4 | 4 | 4 | 4 | 2 | 2 |
| Erythromycin | 0.25 | 32 | 16 | 16 | 16 | 32 | 32 |
| Lincomycin | 0.5 | 32 | 32 | 64 | 32 | 128 | 64 |
| Oxacillin | 0.5 | 16 | 16 | 8 | 8 | 32 | 16 |
| Tetracycline | 8 | 16 | 16 | 16 | 16 | 8 | 8 |

## Mutations observed in the Quinolone-resistance determining regions (QRDRs)

Target sequencing of QRDRs indicated that there were some mutations in both *grlA* (S80F) and *gyrB* (T451S and/or R450S) (Table 4), among those, mutations of both genes were found in CIP- and OFL-1 while only one mutation in *gyrB* (T451S) was found in LEV-1. Additional sequencing of the *S. aureus* strains at days 4th, 6th, 8th, 10th of CIP-, LEV- and OFL-exposure revealed the *grlA* mutation (S80F) to appeared in earlier steps than the ones in *gyrB*, suggesting the primary role of *grlA* mutation in FQ resistant phenotype (Table 4).

## Overexpression of alternative sigma factors in FQ-exposed strains

The data in Fig 2 illustrates that there were upregulations of *sigB* and *sigS* genes in most FQ-exposed strains, except for OFL-2 and LEV-2. The expression of *sigB* and *sigS* genes were recorded highest in LEV-1, at a mean fold change expression of 2.39 and 3.80 respectively, while the most significant downregulation of both genes was recorded in OFL-2 which were about 0.41 and 0.55 fold decrease. In terms of the *S. aureus*-2 group, the upregulation of both genes in CIP-2 made this the only strain in the group that experienced a rise in gene expression of both alternative sigma factors, $\sigma^B$ and $\sigma^S$ in comparison with initial *S. aureus*. In the statistical analysis between two groups of *S. aureus*-1 and *S. aureus*-2 for the same antibiotic, it can be seen that there was a decrease in gene expression in both *sigB* and *sigS* genes of *S. aureus*-2 strains compared with *S. aureus*-1 strains.

## Efflux expression increased in FQ-exposed strains

The effect of FQ exposure on the expression level of the efflux pump genes was checked and compared to the initial *S. aureus*. As shown in Fig 3, all *S. aureus*-1 and 2 strains increased the expression of *mgrA*, *norA*, *norB*, and *norC*. Regarding to the expression level of *mgrA*, which is

**Table 3. The list of single nucleotide polymorphisms (SNPs), variants, and amino acid changes found in *S. aureus*-1 (CIP-1, OFL-1, and LEV-1) and *S. aureus*-2 (CIP-2, OFL-2, and LEV-2) strains but not in *S. aureus* ATCC 29213.**

| SNP/variant and GenBank Accession | Amino acid changes | Systematic and trivial gene name | Protein encoded | Mutations in samples | | | | | | |
|---|---|---|---|---|---|---|---|---|---|---|
| | | | | *S. aureus* ATCC 29213 | CIP-1 | OFL-1 | LEV-1 | CIP-2 | OFL-2 | LEV-2 |
| **G5009GAAC MOPB01000035.1** | **S33_S34insC** | **BJI72_1850 *rimI*** | **Alanine acetyltransferase** | . | √ | √ | √ | √ | √ | √ |
| **TATCTAGATGGATG4318T MOPB01000016.1** | **G309_T315delinsC** | **BJI72_0480 SACOL0573** | **PIN domain containing protein** | . | √ | √ | √ | √ | √ | √ |
| **A13766G MOPB01000012.1** | **No annotated protein** | - | - | . | √ | √ | √ | √ | √ | √ |
| **A13775T MOPB01000012.1** | **No annotated protein** | - | - | . | √ | √ | √ | √ | √ | √ |
| **A13777C MOPB01000012.1** | **No annotated protein** | - | - | . | √ | √ | √ | √ | √ | √ |
| **A13778T MOPB01000012.1** | **No annotated protein** | - | - | . | √ | √ | √ | √ | √ | √ |
| **T13781A MOPB01000012.1** | **No annotated protein** | - | - | . | √ | √ | √ | √ | √ | √ |
| **T13785G MOPB01000012.1** | **No annotated protein** | - | - | . | √ | √ | √ | √ | √ | √ |
| **CG13787C MOPB01000012.1** | **No annotated protein** | - | - | . | √ | √ | √ | √ | √ | √ |
| **C32411A MOPB01000008.1** | **Synonymous Mutation** | **BJI72_0464 *hslO*** | **heat shock protein Hsp33** | . | √ | √ | √ | √ | √ | √ |
| CATAGGCTTGTT22457C MOPB01000021.1 | N3181delinsR | BJI72_1235 Ebh | Hyperosmolarity resistance protein Ebh | . | . | √ | √ | . | √ | √ |
| T86957C MOPB01000021.1 | Synonymous Mutation | BJI72_1284 lpl | Lipoprotein | . | . | √ | √ | . | √ | √ |
| CTT26104C MOPB01000037.1 | K1283delinsRG | BJI72_1954 *fmtB* | methicillin resistance protein FmtB | . | . | √ | √ | . | √ | √ |
| C93466T MOPB01000004.1 | A318T | BJI72_0139 ThlA | acetyl-CoA acetyltransferase | . | √ | . | . | √ | . | . |
| **G308933A MOPB01000004.1** | **R570H** | **BJI72_0342 *grlA*** | **DNA topoisomerase IV subunit A** | . | √ | √ | √ | √ | √ | √ |
| T13964TTA MOPB01000008.1 | V69delinsIRINQ | BJI72_0448 purR | pur operon repressor | . | . | . | . | . | √ | √ |
| G52A MOPB01000047.1 | Synonymous mutation | BJI72_2594 sdrE | serine-aspartate repeat-containing protein E | . | √ | √ | √ | √ | √ | . |

Bolded entries represent SNP, variants, and amino acid changes that appeared in all *S. aureus*-1 and 2 strains, but not in *S. aureus* ATCC 29213.

√, the mutation present; ., no mutation.

involved in the efflux pump regulation of *S. aureus*, LEV-1 exhibited the highest expression level (about 12.63 folds change), followed by CIP-1 and LEV-2 (Fig 3A). Besides, *norA* was also upregulated in all FQ-exposed strains, especially in CIP-1 and CIP-2 of which the expression increased by about 57.61 and 70.78 folds respectively compared to initial strains (Fig 3B). Among *S. aureus*-1 strains, OFL-1 showed the lowest expression in *mgrA*, *norA*, *norB*, and *norC*. A similar result was also found in OFL-2 when compared with the remaining *S. aureus*-2 strains. In addition, almost all strains belonging to *S. aureus*-1 group had higher expression in *mgrA*, *norA*, *norB*, and *norC* compared to the strains which were treated with the same antibiotic in *S. aureus*-2 group.

**Table 4. Sequencing result of *grlA* and *gyrB* of *S. aureus* strains.**

| Strains | MIC (mg/L) | Appearance of mutation in | |
|---|---|---|---|
| | | *grlA* | *gyrB* |
| **Ciprofloxacin** | | | |
| *S. aureus* ATCC 29213 | 0.125 | - | - |
| CIP[4]-resistant | 2 | - | - |
| CIP[6]-resistant | 8 | S80F | - |
| CIP[8]-resistant | 8 | S80F | - |
| CIP[10]-resistant | 16 | S80F | T451S |
| CIP[12]-resistant-1 (CIP-1) | 16 | S80F | R450S and T451S |
| CIP-resistant-2 (CIP-2) | 4 | - | T451S |
| **Ofloxacin** | | | |
| *S. aureus* ATCC 29213 | 0.25 | - | - |
| OFL[4]-resistant | 2 | - | - |
| OFL[6]-resistant | 2 | - | - |
| OFL[8]-resistant | 4 | S80F | - |
| OFL[10]-resistant | 8 | S80F | - |
| OFL[12]- resistant-1 (OFL-1) | 8 | S80F | T451S |
| OFL- resistant-2 (OFL-2) | 4 | - | - |
| **Levofloxacin** | | | |
| *S. aureus* ATCC 29213 | 0.125 | - | - |
| LEV[4]-resistant | 0.5 | - | - |
| LEV[6]-resistant | 1 | - | - |
| LEV[8]-resistant | 2 | - | - |
| LEV[10]-resistant | 4 | - | - |
| LEV[12]- resistant-1 (LEV-1) | 4 | A64A | T451S |
| LEV- resistant-2 (LEV-2) | 2 | - | T451S |

No mutation was observed in *gyrA* and *grlB*. The substitution mutations resulted from changes in the following nucleotides: S80F, UCC to UUC; A64A, GCG to GCA; R450S, AGA to AGU; T451S, ACG to UCG.

-, no mutation observed.

[4, 6, 8, 10, 12], *S. aureus* strains at day 4th, 6th, 8th, 10th, and 12th (CIP-, OFL- and LEV-resistant *S. aureus*-1) of CIP-, OFL- and LEV-exposure.

## Overexpression of genes related to FQ-resistant development in *S. aureus*

According to the WGS results, there were genomic mutations located in the sequence of *rimI* and *fmtB* in both *S. aureus*-1 and -2 strains. Besides, 7 SNPs were also found in all FQ-exposed strains which were located in the upstream region of *norA* and the gene encoding hypothetical protein BJI72_0645 (Table 3). The results suggested that these variants might be related to FQ-resistant development in *S. aureus*. Therefore, investigating the expression of *rimI*, *fmtB*, and the gene encoding hypothetical protein BJI72_0645 should be carried out to better understand the relationship between these mutations and the FQ-resistant development in *S. aureus*.

Under the effects of FQ exposure, *rimI* was overexpressed in all FQ-exposed strains, especially the expression increased by 30.43 folds in LEV-1 (Fig 4A). The data also indicated that *fmtB* was upregulated in *fmtB*-mutant strains (Fig 4B). Besides, although there was no *fmtB* mutation in CIP-1 and CIP-2 which were resistant to ciprofloxacin, *fmtB* was also overexpressed in both strains. Regarding the expression level of the gene encoding hypothetical protein BJI72_0645, this gene was also overexpressed in all FQ-exposed strains, especially LEV-1

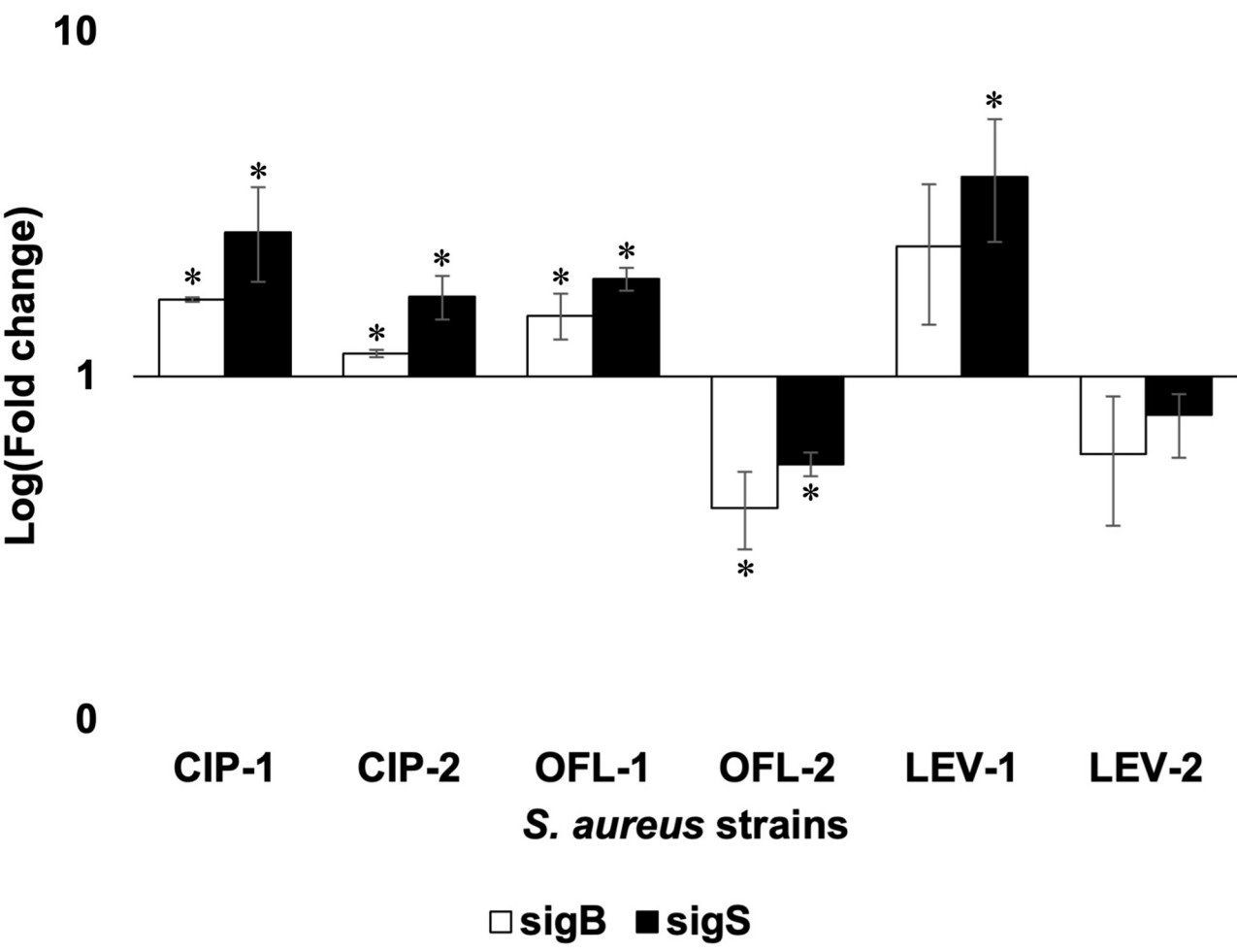

**Fig 2. RT-qPCR analysis of *sigB* and *sigS* in *S. aureus* ATCC 29213 and its exposed strains.** * indicated a significant difference in gene expression between initial *S. aureus* and FQ-exposed *S. aureus* strains.

(Fig 4C). In addition, this study also found that the expression of *rimI*, *fmtB*, and the gene encoding hypothetical protein BJI72_0645 in *S. aureus*-1 strains was higher than that in corresponding *S. aureus*-2 strains in all cases.

## Discussion

### Sub-MIC exposure to FQ altered the antibiotic susceptibility profile of *S. aureus*

After 12 days of exposure to the sub-MICs of FQs, *S. aureus* increased its MIC values and became resistant to exposed antibiotics. Serial exposure to sub-MICs can create positive selection pressure that drove the development of resistant phenotypes, which is in agreement with previous literature [48–50]. Although *S. aureus* had a decrease in MIC value after 10 days of culturing in an antibiotic-free medium, indicating some reversion of resistance trait, *S. aureus*-2 strains kept their resistance to FQs. Besides, after being serial exposed to CIP, OFL, and LEV, *S. aureus* became resistant not only to the exposed antibiotics and other FQs but also to

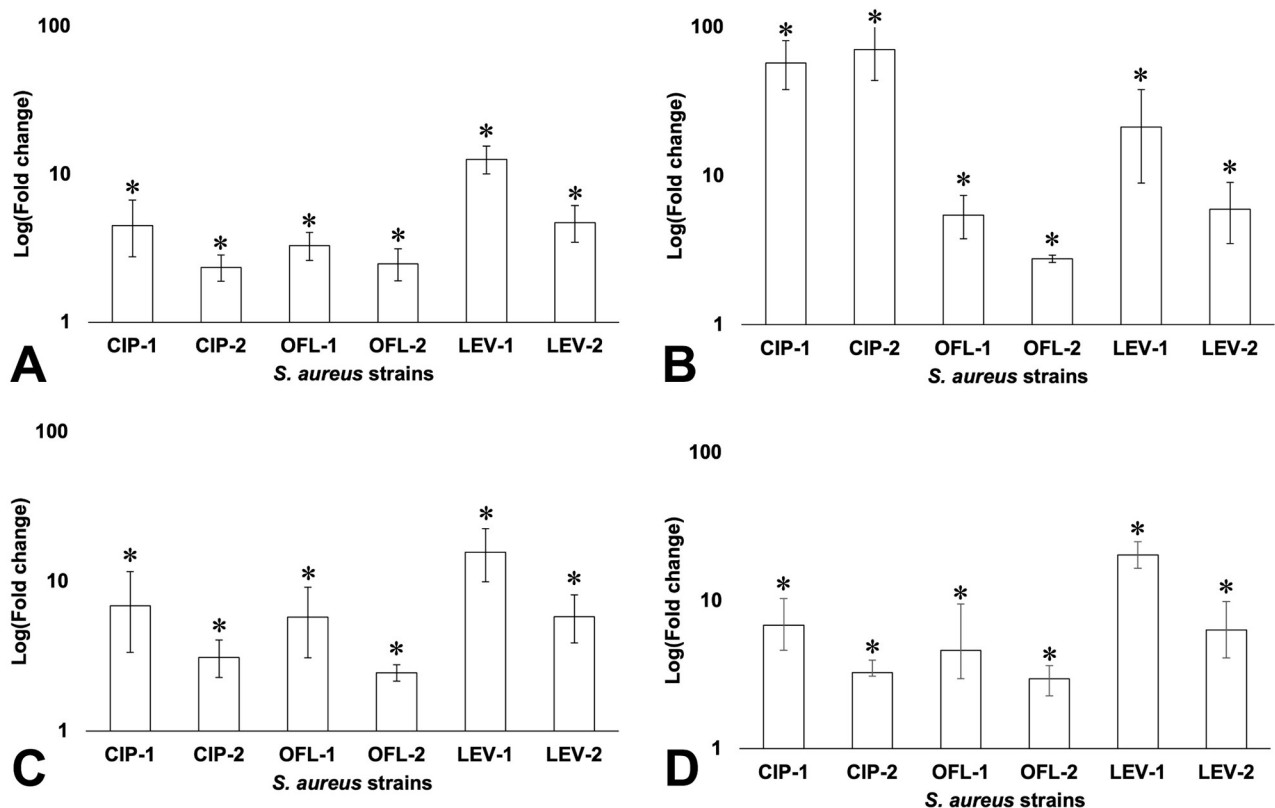

**Fig 3. RT-qPCR analysis of *mgrA* (A) and multidrug efflux pump *norA* (B), *norB* (C), and *norC* (D) in *S. aureus* ATCC 29213 and its exposed strains.** * indicated a significant difference in gene expression between initial *S. aureus* and FQ-exposed *S. aureus* strains.

unrelated antibiotics such as ampicillin, amoxicillin, doxycycline, erythromycin, and lincomycin. The results suggested that serial FQ exposure resulted in cross-resistance to unrelated antibiotics and the emergence of multidrug-resistant phenotypes in *S. aureus*. It should be well noted that this cross-resistance development under CIP exposure was well observed in Gram-positive bacteria such as *Actinobacillus pleuropneumoniae* [51], *S. aureus* [52], *Mycobacterium tuberculosis* [53]; but not in Gram-negative, ones like *Salmonella enterica* [54]; *Pseudomonas aeruginosa* [55].

### Genomic alterations involved in FQ-resistant development in *S. aureus*

It has been shown that the acquisition of FQ resistance is mainly due to the mutations in target enzymes, topoisomerase IV (GrlA/B) as well as DNA gyrase (GyrA/B) [56]. In *S. aureus*, the mutations occurred more frequently in QRDRs of *grlA* (S80F or Y, E84K, and A116E or P) and *gyrA* (S84L or A, S85P, and E88K) which were described as the primary FQ resistance mechanism [56]. Other studies have found that the mutations in QRDRs of *gyrB* including D437N, R458E, D432N, and N470D also contributed to FQ resistance [57, 58].

In our *in vitro*-induced FQ-resistant model, we found mutations in both *grlA* (S80F) and *gyrB* (T451S and/or R450S), among those, mutations of both genes were found in CIP- and OFL-1 while only one mutation in *gyrB* (T451S) was found in LEV-1. Additional sequencing of the *S. aureus* strains at days 4th, 6th, 8th, 10th of CIP-, LEV- and OFL-exposure revealed the *grlA* mutation (S80F) to appeared in earlier steps than the ones in *gyrB*, suggesting the primary

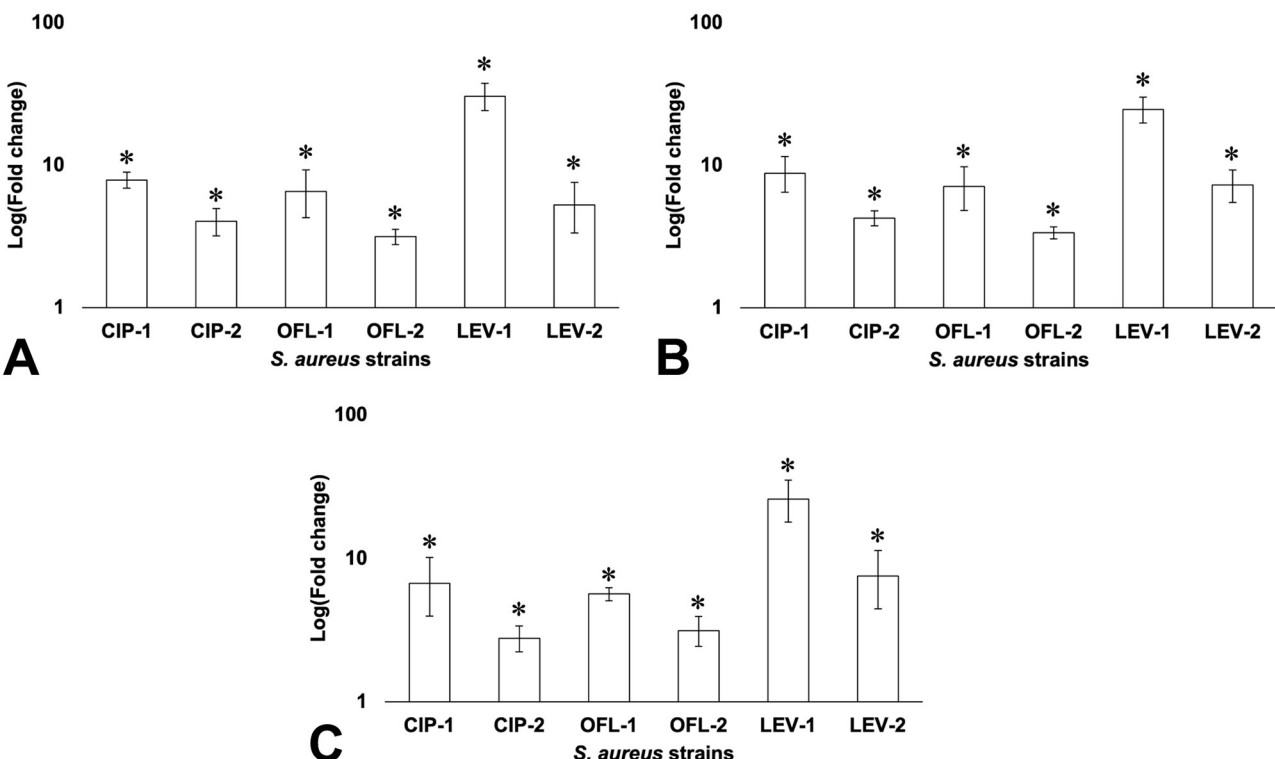

**Fig 4. RT-qPCR analysis of *rimI* (A), *fmtB* (B), and the gene encoding hypothetical protein BJI72_0645 (C) in *S. aureus* ATCC 29213 and its exposed strains.** * indicated a significant difference in gene expression between initial *S. aureus* and FQ-exposed *S. aureus* strains.

role of *grlA* mutation in FQ resistant phenotype. These findings suggested that the mutation in *grlA* (S80F) primarily induced the emergence of FQ resistance and might support the mutations in *gyrB* (T451S); which was required for high-level FQ resistance in *S. aureus*.

In addition, we observed that mutations in both *grlA* and *gyrB* occurred in all *S. aureus*-1 strains. These mutations resulted in amino acid alterations in both DNA gyrase and topoisomerase IV, which were FQ targets. Consequently, that led to the emergence of FQ resistance in *S. aureus*-1 due to the alteration of binding sites between FQ and target enzymes. However, interestingly, all mutations in *grlA* disappeared in *S. aureus*-2 without a significant MIC decrease in LEV-2 and only a four-time reduction in CIP-2. With OFL-2, it is impressive that no mutation in *gyrA*, *gyrB*, *grlA*, and *grlB* QRDRs was retained while it kept most of its resistant ability. These data suggested that mutations in FQ targets are important in the presence of FQs but they may not be essential mechanisms to mediate FQ and multidrug-resistant phenotype. One previous study on *Enterococcus faecium* also showed no mutation detected in QRDRs of *gyrA*, *gyrB*, *parC*, and *parE* genes in the strain selected from 2-step FQ-resistant mutant induction [59], suggesting the importance of other mechanisms.

### Changes in expression of alternative sigma factors associated with the adaptive response of *S. aureus* under serial FQ exposure

Alternative sigma factors (σ) are an essential component of core RNA polymerase (RNAP) that helps determine promoter selectivity [60]. The association of appropriate alternative sigma factors with core RNAP can help bacteria perform specialized cellular functions or stress-adaptive

responses through redirection of transcription initiation [61]. The stress-response alternative sigma factors σ[B] and σ[S] contribute to bacterial survival under multiple stresses. σ[B] affects cellular processes including oxidative stress resistance, drug resistance, stress adaptation, and biofilm formation [62]. Conversely, σ[S] is activated in response to stresses such as stress resistance, DNA and cell wall damage, cell morphology alterations, metabolism, virulence, and lysis [63].

In this study, the transcriptional analysis of *sigB* and *sigS* genes showed that both genes increased their expression in *S. aureus*-1 strains in responding to FQ exposure but reduced in the *S. aureus*-2 strains when FQs were withdrawn. It is understandable as sigma factors reacted quickly to extracytoplasmic stresses in *S. aureus* in which under standard conditions, sigma factor transcription remains low but can be upregulated in response to external stimuli, especially chemicals leading to DNA damage and cell wall disruption such as antibiotics [47]. In addition, sigma factors enable bacteria to rapidly detect antimicrobial signals, activate resistance genes, and repair mechanisms specific to the antimicrobials [64]. In our case, both *sigB* and *sigS* seemed to play an important role in *S. aureus* fitness and adaptation to FQs. It could be the reason for the marked change in proteomic profiles of the bacteria in responding to FQs [65].

## Sub-MIC exposure to FQs led to overexpression of the efflux pump and its regulator

Several specific efflux pumps have been associated with antibiotic resistance in clinical isolates of *S. aureus*. With respect to FQ, *norA*, *norB*, and *norC* are the most important efflux pumps localized on the cytoplasmic membrane of *S. aureus* [45]. These pumps can extrude many chemically and structurally dissimilar compounds, namely hydrophilic and hydrophilic FQs (such as norfloxacin, ciprofloxacin, moxifloxacin, and garenoxacin), dyes (like ethidium bromide and rhodamine) and biocides (tetraphenylphosphonium and cetrimide) [37, 66]. It has been shown that *mgrA* functions as a positive regulator of *norB* and a negative regulator of *norA* and *norC* [67].

Regarding the expression level of *mgrA*, the results in this study were consistent with our previous study on proteomic analysis [65]. In order to test the effects of overexpression of *mgrA* on multidrug efflux pump *norA*, *norB*, and *norC*, RT-qPCR was applied for determining their expression. In analysis, the expression of *norA*, *norB*, and *norC* was increased in all tested strains. It means that efflux pumps were activated and could be one of the mechanisms leading to the occurrence of multidrug-resistant *S. aureus* strains. Our results are in agreement with the previous study, in which overexpression of MgrA has led to a change in the expression level of *norA* and *norB* [66]. However, the increased expression of *norC* in this study was not associated with the regulation of *mgrA*, and it might result from mutational alterations in uncharacterized loci that affect the expression of this gene.

The decrease in efflux expression in *S. aureus*-2 compared to those of *S. aureus*-1 suggests that under the FQ stressor in the environment, the efflux activity of *S. aureus* was promoted as an adaptive response to resist the antibiotics. However, the efflux activity could decrease when *S. aureus*-1 was cultured continuously in antibiotic-free media even though it still kept its resistance to FQ. This is consistent with the result that *S. aureus* turned to an FQ-resistant phenotype after 12-day serial exposure to FQ, and the MIC values witnessed a slight decrease after 10-day continuous culturing *S. aureus*-1 in antibiotic-free media.

## Sub-MIC exposure to FQs affected the protein acetylation and multi-drug resistant phenotype of *S. aureus*

Protein acetylation is one of the major post-translational modifications (PTMs) which play an important role in cell signaling and occurs when the cell encounters specific environmental

stress conditions [68–70]. RimI encoded by *rimI* is a ribosomal protein S18 acetyltransferase which catalyzes the acetylation of the N-terminal alanine of ribosomal protein S18; and also acts as an N-epsilon-lysine acetyltransferase that catalyzes acetylation of several proteins [71, 72]. In this study, an insertion mutation (S33_S34insC) in *rimI* was found in all *S. aureus*-1 and 2 that might affect the RimI activities and the protein acetylation in *S. aureus*. In terms of expression analysis, *rimI* was overexpressed in these mutant strains compared to the initial *S. aureus*. These results suggested that the mutation in *rimI*, as well as its overexpression in FQ-exposed *S. aureus*, was an adaptive response under antibiotic stressors. This might be associated with acetylation which alters mRNA translation efficiency in several proteins in *S. aureus* to maintain its survival in an antibiotic-stress environment.

The *fmtB* gene codes for FmtB, a ~263 kDa cell wall-anchored protein [73]. Although there is limited information about the function of *fmtB*, it has been proven to contribute to the methicillin-resistant phenotype in *S. aureus* [74, 75]. According to the antibiotic susceptibility profile of seven *S. aureus* strains, after being exposed to FQ, *S. aureus* enhanced its resistance to FQ and other antibiotics including B-lactam. Besides, the mutation K1283delinsRG (CTT26104C) in *fmtB* sequence was also found in OFL- and LEV-exposed strains. For RT-qPCR analysis, *fmtB* was upregulated in all mutant strains. Although there was no mutation on *fmtB* found in CIP-1 and CIP-2, which were resistant to ciprofloxacin, *fmtB* was also overexpressed in both strains. It was suggested that the *in vitro*-induced FQ resistance cooperating with the acetylation modification somehow affects the *fmtB* expression and the antibiotic resistance of *S. aureus* to FQs and other antibiotics of unrelated groups leading to a multi-drug resistance phenotype.

According to whole genome analysis, 7 SNPs were found in the upstream region of both *norA* and the gene encoding hypothetical protein BJI72_0645. The RT-qPCR results indicated that the variants might play a crucial role in regulating the transcription of these genes because both were overexpressed in all FQ-exposed *S. aureus*.

## From genetic alterations to protein expression

The development of antibiotic resistance might involve not only transcriptional regulation and genetic modifications but also translational regulation. In the proteomic study of our research group, under the FQ stressor in the environment, there were 147 unique proteins in *S. aureus*-1 and *S. aureus*-2 which changed their expression in comparison to the initial strain [65]. Regarding the molecular function of differently expressed proteins, 93 proteins were responsible for binding, 89 proteins for catalytic activity, 28 proteins for the structural constituent of ribosome, and 10 other proteins were involved in transcription factor activity (6), transporter activity (2) and antioxidant activity (2). Proteins involved in SOS/stress response and antibiotic resistance and pathogenesis were upregulated upon FQ exposure. Among them, RecA and MgrA are global regulators which are strongly implicated in antibiotic resistance development [76, 77]. The presence of RecA leads to the upregulation of SOS genes which involve in DNA repair and promote the autoproteolysis cleavage of LexA (an SOS gene repressor) [78, 79]; while MgrA affects multiple *S. aureus* genes which encoded proteins involving multidrug resistance, autolytic activity, and virulence [80, 81]. The fluctuation in protein expression depends on many factors including extracellular agents such as harsh temperatures, chemical and antibiotic stress in the environment, and intracellular agents such as modifications in the genetic materials. Previous studies have proven that the elements belonging to genetic code including amino acid [82], untranslated regions [83–86], length [86], GC content [87–89], and mRNA secondary structure [90] affected the regulation of protein expression. However, under the serial exposure of *S. aureus* to FQs, the direct relationship between protein expression and

genetic modifications found in this study was not proved. They only suggested that resistance to multiple drugs including FQs under drug exposure generally requires the contribution of multiple proteins and processes. In short, the genetic mutations found in FQ-exposed *S. aureus* and overexpression of global regulators may be the key to the expression change of a range of proteins which assist this bacterium adapt the antibiotic-containing environments.

## Conclusion

Exposure to sub-MICs of FQs provided positive selection pressure for the development of resistance traits leading to alterations to the antibiotic susceptibility profile and transcription of the pathogen. These alterations were reflected in the DNA mutations and the change in mRNA expression levels of efflux pumps, especially alternative sigma factors. Multidrug resistance or antibiotic-resistant development is a complicated and multifactorial process. The found mutations might be spontaneous, but their combination is clearly a cause of the resistance status observed in the bacterium.

FQs presently play a vital role in saving lives and are employed in treating a broad spectrum of infectious diseases or even cancers. Nonetheless, the extensive utilization of FQs in human and animal healthcare has led to a growing number of antibiotic-resistant pathogens. The ease of *S. aureus* to develop FQ resistance emphazised the prudent and appropriate use of antibiotics in preventing the selection of resistant bacteria. Additionally, molecules with ability to interfere the resistance development would be highly recommended in FQ combination therapy.

## Supporting information

**S1 Fig. Rep-PCR fingerprints of initial *S. aureus* ATCC 29213 and FQ-exposed strains.** Ladder 100 bps; 1, *S. aureus* ATCC 29213; 2, CIP-1; 3, CIP-2; 4, LEV-1; 5, LEV-2; 6, OFL-1; 7, OFL-2.
(TIF)

**S1 Table. The effect of reserpine on antibiotic susceptibility profile of initial and *S. aureus*-1 strains.**
(DOCX)

**S2 Table. List of single nucleotide polymorphisms (SNPs), variants, and amino acid changes found in *S. aureus*-1 (CIP-1, OFL-1, and LEV-1) and *S. aureus*-2 (CIP-2, OFL-2, and LEV-2) strains but not in *S. aureus* ATCC 29213.**
(DOCX)

**S3 Table. The statistical analysis of fold change in gene expression between initial *S. aureus* strain and FQ-exposed *S. aureus* strains (mean ± standard error).**
(DOCX)

**S1 File. MIC values of *S. aureus* during sub-MIC exposure to FQs including ciprofloxacin, ofloxacin, or levofloxacin.**
(DOCX)

## Acknowledgments

The study was supported by The Youth Incubator for Science and Technology Programme, managed by Youth Promotion Science and Technology Center—Ho Chi Minh Communist

Youth Union and Department of Science and Technology of Ho Chi Minh City under the contract number 34/2022/ HĐ-KHCNT-VU.

## Author Contributions

**Conceptualization:** Thi Thu Hoai Nguyen.

**Data curation:** Thuc Quyen Huynh, Van Nhi Tran, Van Chi Thai, Hoang An Nguyen, Ngoc Thuy Giang Nguyen, Minh Khang Tran, Thi Phuong Truc Nguyen, Cat Anh Le, Le Thanh Ngan Ho, Navenaah Udaya Surian, Swaine Chen.

**Formal analysis:** Thuc Quyen Huynh, Van Nhi Tran, Hoang An Nguyen, Ngoc Thuy Giang Nguyen, Navenaah Udaya Surian, Swaine Chen, Thi Thu Hoai Nguyen.

**Funding acquisition:** Thuc Quyen Huynh, Thi Thu Hoai Nguyen.

**Investigation:** Thuc Quyen Huynh, Van Nhi Tran, Minh Khang Tran, Thi Phuong Truc Nguyen, Cat Anh Le, Navenaah Udaya Surian, Swaine Chen, Thi Thu Hoai Nguyen.

**Methodology:** Swaine Chen, Thi Thu Hoai Nguyen.

**Project administration:** Thi Thu Hoai Nguyen.

**Resources:** Thi Thu Hoai Nguyen.

**Software:** Thi Thu Hoai Nguyen.

**Supervision:** Thi Thu Hoai Nguyen.

**Visualization:** Thi Thu Hoai Nguyen.

**Writing – review & editing:** Thi Thu Hoai Nguyen.

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
