## [Decision Letter · Decision Letter 0]

3 May 2023

PONE-D-23-07124Genomic alterations involved in fluoroquinolone-resistant development of Staphylococcus aureusPLOS ONE

Dear Dr. Nguyen,

Thank you for submitting your manuscript to PLOS ONE. After careful consideration, we feel that it has merit but does not fully meet PLOS ONE’s publication criteria as it currently stands. Therefore, we invite you to submit a revised version of the manuscript that addresses the points raised during the review process.

We look forward to receiving your revised manuscript.

Kind regards,

Farah Al-Marzooq, MD, PhD

Academic Editor

PLOS ONE

Journal Requirements:

When you resubmit, please ensure that you provide the correct grant numbers for the awards you received for your study in the ‘Funding Information’ section."

Additional Editor Comments:

Please revise the manuscript as advised by the reviewer.

Reviewers' comments:

Reviewer's Responses to Questions

**Comments to the Author**

1. Is the manuscript technically sound, and do the data support the conclusions?

Reviewer #1: Yes

2. Has the statistical analysis been performed appropriately and rigorously? 

Reviewer #1: Yes

3. Have the authors made all data underlying the findings in their manuscript fully available?

Reviewer #1: Yes

4. Is the manuscript presented in an intelligible fashion and written in standard English?

Reviewer #1: Yes

5. Review Comments to the Author

Reviewer #1: The aim of this work is interesting and the results convincing. The idea reported is remarkable and the paper is well done. The Just a few suggestions to improve the readability of the text helping the reader to better understand the paper.

Reviewer Comments:

-Abstract: Abstract is too lon. It must be not more than 250 words. Please add background/problem statement and objectives. The abstract did not reflect the main findings of the research.

-Introduction: It is good in story line however more information need to be addressed what cause the resistance to the antibiotic and why mutation occurred or what trigger this mutation in normal circumstances not mentioned. Mention the type of mutations and type of important drug efflux pumps of S. aureus belonging families which contributes to resistance to fluoroquinolones? Also introduction must be explain the gene encode for multidrug resistance efflux pump

-Why choose (grlA/B) not (parC/parE)? this is not address in Introduction section as well. What's the total length of grlA/B?

-Methodology: comprehensive and adequate, but you can add the section of statistical analysis in methods; the authors may use correlation to be more informative.

-Why used S. aureus ATCC 29213 and expose it FQ to obtain resistance strains? you could isolate resistance strains to FQ from different clinical settings what the purpose?

-Results:

-in line 128 Comparison between strains before exposure to treatment with FQ and after insufficient should be reformulated in a more understandable way

-Table 2, the title and the content are not well described in the paper, please change the title or add some sentence or reference in the paper.

-Why did the authors focus on grlA/B not parC and mutation in this gene first step in the resistance to most fluoroquinolones.

-Fig 1 hardly to read. Only fig quality is not enough?

-Quality of images was low?

Discussion

Interestingly, we observed the 277 revertance of all mutations in grlA (turning back to the wild-type genotype) without a significant MIC 278 decrease in OFL- and LEV-2 and only a four-time reduction in CIP-revertant strain when both mutations 279 in grlA (S80F) and in gyrB (R450S) were together disappeared. Moreover, even though OFL-2 kept most 280 of its resistant ability, no mutation in gyrA, gyrB, grlA, and grlB QRDRs was retained. Our results were 281 similar to the observation in E. faecium of Yoshihiro et al. who proposed that other resistance mechanism(s) 282 such as multidrug resistance efflux pumps might be involved in the development of FQ resistance. How similar"?/ Statistical analysis is needed.

There is a long-established correlation between increased resistance and fitness cost [30]. If the FQ-resistance mechanism imposes a fitness cost, it would be advantageous for bacteria to be able to exhibit the resistance phenotype only when the threat exists. Even if there is no fitness cost, the production of resistance-mediating proteins could impose an unnecessary metabolic burden on the bacteria in an 289 antimicrobial-free environment. Rewrite the sentences in a more understandable way?

conclusion

Author must address whether resistance to FQ is an alarming situation or not since the resistances of s. aureus to FQ were reported in many other countries as well.

References:

Many references are OLD. Please include the latest especially on Introduction and Discussions. Thus, we prefer to use the update and following references:-

1-Hussein, RA, Al-Ouqaili MTS, Majeed YH. (2022). Detection of clarithromycin resistance and 23SrRNA point mutations in clinical isolates of Helicobacter pylori isolates: Phenotypic and molecular methods, Saudi Journal of Biological Sciences, 29 (1).

2- Al-Ouqaili, M.T.S., Al-Kubaisy, S.H.M., Al-Ani, N.F.I. Biofilm antimicrobial susceptibility pattern for selected antimicrobial agents against planktonic and sessile cells of clinical isolates of staphylococci using MICs, BICs and MBECs. Asian Journal of Pharmaceutics, volume 12, Issue 4, October-December 2018, Pages S1375-S1383.

3- Al-Qaysi, A. K., Al-Ouqaili, . M. T. & Al-Meani, . S. A. (2020). Ciprofloxacin- and gentamicin-mediated inhibition of Pseudomonas aeruginosa biofilms is enhanced when combined the volatile oil from Eucalyptus camaldulensis. SRP, 11 (7), 98-105.

4-Chinemerem Nwobodo D, Ugwu MC, Oliseloke Anie C, Al-Ouqaili MTS, Chinedu Ikem J, Victor Chigozie U, Saki M. Antibiotic resistance: The challenges and some emerging strategies for tackling a global menace. J Clin Lab Anal. 2022 Sep;36(9):e24655. doi: 10.1002/jcla.24655. Epub 2022 Aug 10. PMID: 35949048; PMCID: PMC9459344.

6. PLOS authors have the option to publish the peer review history of their article (what does this mean?). If published, this will include your full peer review and any attached files.

Reviewer #1: **Yes: **Rawaa A. Hussein

---

## [Author Response · Author response to Decision Letter 0]

14 Jun 2023

Dear Reviewers, 

We have attached a file "response to reviewers" in the re-submitted files where we answered all your comments one by one.

We thank you very much for your time and valuable comments. They are very helpful to improve the quality of the manuscript.

With best regards,

---

## [Decision Letter · Decision Letter 1]

19 Jun 2023

Genomic alterations involved in fluoroquinolone-resistant development of Staphylococcus aureus

PONE-D-23-07124R1

Dear Dr. Nguyen,

We’re pleased to inform you that your manuscript has been judged scientifically suitable for publication and will be formally accepted for publication once it meets all outstanding technical requirements.

Kind regards,

Farah Al-Marzooq, MD, PhD

Academic Editor

PLOS ONE

Additional Editor Comments (optional):

Just a minor comment on the title , change resistant to resistance , as it is more suitable

Reviewers' comments:

Reviewer's Responses to Questions

**Comments to the Author**

1. If the authors have adequately addressed your comments raised in a previous round of review and you feel that this manuscript is now acceptable for publication, you may indicate that here to bypass the “Comments to the Author” section, enter your conflict of interest statement in the “Confidential to Editor” section, and submit your "Accept" recommendation.

Reviewer #1: All comments have been addressed

2. Is the manuscript technically sound, and do the data support the conclusions?

Reviewer #1: Yes

3. Has the statistical analysis been performed appropriately and rigorously? 

Reviewer #1: Yes

4. Have the authors made all data underlying the findings in their manuscript fully available?

Reviewer #1: Yes

5. Is the manuscript presented in an intelligible fashion and written in standard English?

Reviewer #1: Yes

6. Review Comments to the Author

Reviewer #1: No additional comments for author,including concerns about dual publication,research ethics, or publication ethics

It’s interesting work

7. PLOS authors have the option to publish the peer review history of their article (what does this mean?). If published, this will include your full peer review and any attached files.

Reviewer #1: **Yes: **Rawaa A. Hussein

---

## [Editor Report · Acceptance letter]

17 Jul 2023

PONE-D-23-07124R1 

Genomic alterations involved in fluoroquinolone resistance development in *Staphylococcus aureus*

Dear Dr. Nguyen:

I'm pleased to inform you that your manuscript has been deemed suitable for publication in PLOS ONE. Congratulations! Your manuscript is now with our production department. 

Kind regards, 

on behalf of

Dr. Farah Al-Marzooq 

Academic Editor

PLOS ONE